# External quality assessment of Rift Valley fever diagnosis in countries at risk of the disease: African, Indian Ocean and Middle-East regions

Aurélie Pedarrieu[1,2], Fatiha El Mellouli[3], Hanane Khallouki[3], Khalil Zro[4], Ghizlane Sebbar[4], Soufien Sghaier[5], Hafsa Madani[6], Nadera Bouayed[6], Modou Moustapha Lo[7], Mariame Diop[7], Ahmed Bezeid Ould El Mamy[8], Yahya Barry[8], Marthin Dakouo[9], Abdallah Traore[9], Haladou Gagara[10], Maman Moutari Souley[10], Sara Acha[11], Laurenco Mapaco[11], Jelly Chang'a[12], Denis Nyakilinga[12], Baratang A. Lubisi[13], Thabisile Tshabalala[13], Claudia Filippone[14], Jean Michel Heraud📷[14], Sitty-Bahyat Chamassy[15], Abdou Achiraffi[15], Nicolas Keck📷[16], Gilda Grard[17,18], Kareem Abdelfattah Abdelwahab Mohammed[19], Abdulwahed Mohammed Alrizqi[19], Catherine Cetre-Sossah📷[1,20]*

1 ASTRE, Univ Montpellier, CIRAD, INRA, Montpellier, France, 2 CIRAD, UMR ASTRE, F-34398 Montpellier Cedex, France, 3 Laboratoire Régional d'Analyses et de Recherches de Casablanca, Office National de la Sécurité Sanitaire des aliments), Nouaceur, Casablanca, Morocco, 4 Biopharma, Rabat, Morocco, 5 Département de Virologie, Institut de la Recherche Vétérinaire de Tunisie (IRVT), Université de Tunis El Manar, Tunis, Tunisia, 6 Laboratoire Central Vétérinaire d'Alger, Institut National de Médecine Vétérinaire (INMV), Mohammadia, Algeria, 7 Institut Sénégalais de Recherches Agricoles, Laboratoire National de l'Elevage et de Recherches Vétérinaires (ISRA-LNERV), Dakar, Senegal, 8 Office National de Recherches et de Développement de l'Elevage (ONARDEL), Nouakchott, Mauritania, 9 Laboratoire Central Vétérinaire (LCV), Bamako, Mali, 10 Laboratoire Central de l'Elevage (LABOCEL), Niamey, Niger, 11 Agrarian Research Institute of Mozambique, Directorate of Aninal Science, Central Veterinary Laboratory, Maputo, Mozambique, 12 Centre for Infectious Diseases and Biotechnology, Tanzania Veterinary Laboratory Agency, Dar es Salaam, Tanzania, 13 Agricultural Research Council-Onderstepoort Veterinary Research (ARC-OVR), Onderstepoort, South Africa, 14 Institut Pasteur de Madagascar, Unité de Virologie, Antananarivo, Madagascar, 15 Laboratoire vétérinaire et d'analyses départemental (LVAD976), Mayotte, France, 16 Laboratoire Départemental Vétérinaire (LDV34), Montpellier, France, 17 Centre National de Référence sur les arboviruses (CNR Arbovirus), Institut de Recherche Biomédicale des Armées (IRBA), Marseille, France, 18 Unité des Virus Émergents (UVE: Aix-Marseille Univ-IRD 190-Inserm 1207-IHU Méditerranée Infection), Marseille, France, 19 The Ministry of Environment, Water and Agriculture (MEWA), Jazan Veterinary Diagnostic Laboratory, Jizan, Kingdom of Saudi Arabia, 20 CIRAD, UMR ASTRE, F-97490 Sainte-Clotilde, La Réunion, France

* catherine.cetre-sossah@cirad.fr

**Data Availability Statement:** All relevant data are within the manuscript.

**Funding:** This study was partly funded by the SURE project, Préfecture de la Réunion, INTERREG

## Abstract

Rift Valley fever virus (RVFV), an arbovirus belonging to the *Phlebovirus* genus of the *Phenuiviridae* family, causes the zoonotic and mosquito-borne RVF. The virus, which primarily affects livestock (ruminants and camels) and humans, is at the origin of recent major outbreaks across the African continent (Mauritania, Libya, Sudan), and in the South-Western Indian Ocean (SWIO) islands (Mayotte). In order to be better prepared for upcoming outbreaks, to predict its introduction in RVFV unscathed countries, and to run efficient surveillance programmes, the priority is harmonising and improving the diagnostic capacity of endemic countries and/or countries considered to be at risk of RVF. A serological inter-laboratory proficiency test (PT) was implemented to assess the capacity of veterinary laboratories to detect antibodies against RVFV. A total of 18 laboratories in 13 countries in the

FEDER TROI 2015-2017 in the framework of the DP One Health Indian Ocean (www.onehealth-oi. org). There was no additional external funding received for this study.

**Competing interests:** The authors have declared that no competing interests exist.

Middle East, North Africa, South Africa, and the Indian Ocean participated in the initiative. Two commercial kits and two in-house serological assays for the detection of RVFV specific IgG antibodies were tested. Sixteen of the 18 participating laboratories (88.9%) used commercial kits, the analytical performance of test sensitivity and specificity based on the seroneutralisation test considered as the reference was 100%. The results obtained by the laboratories which used the in-house assay were correct in only one of the two criteria (either sensitivity or specificity). In conclusion, most of the laboratories performed well in detecting RVFV specific IgG antibodies and can therefore be considered to be prepared. Three laboratories in three countries need to improve their detection capacities. Our study demonstrates the importance of conducting regular proficiency tests to evaluate the level of preparedness of countries and of building a network of competent laboratories in terms of laboratory diagnosis to better face future emerging diseases in emergency conditions.

# 1 Introduction

Rift Valley fever (RVF), a zoonotic and mosquito-borne disease which primarily affects livestock (ruminants and camels) and humans is caused by an arbovirus belonging to the *Phlebovirus* genus of the *Phenuiviridae* family [1]. The virus was first identified in the Great Rift Valley in Kenya in 1930 [2], then successively throughout the African continent with the first major occurrence of the disease causing mortalities in both humans and livestock in the Nile Delta in Egypt in 1977–1978 [3], then in West Africa, specifically in Mauritania and Senegal in 1987 [4, 5]. The virus has been endemic in sub-Saharan Africa ever since [6, 7]. In 2000, it spread beyond mainland Africa to Saudi Arabia and Yemen [8–10], to the Indian Ocean region (Madagacar) in 1990 and 2008 [11, 12], and to the Comoros archipelago (Mayotte, Union of the Comoros) in 2007 [13, 14].

Outbreaks of RVF are ongoing across the African continent, including in Mauritania [15], Libya [16] and Sudan [17] and across the South-Western Indian Ocean (SWIO) islands, such as the one that occurred in Mayotte in 2018–2019 [18] involving human deaths. RVF is characterised by mass abortion and high mortality rates of neonates in the ruminant population. It continues to pose a threat to neighbouring continents where livestock mobility, immunologically naïve livestock as well as potential mosquito vectors are present [19, 20]. Livestock mobility is one of the main pathways for RVF spread from the African endemic areas to North Africa and the Middle East. Indeed, the risk of virus introduction in RVF free countries is linked to imports of infected animals from endemic areas, through trade or transhumance, or following socio-political conflicts [21]. The deployment of an early warning system adapted to countries at risk in order to avoid the introduction of the virus in a RVF free country or to better control the spread of the virus if the disease is already endemic in the country will require (i) creating an active surveillance system by sampling animal sentinel herds in the season when vector abundance is highest every year, (ii) updating prediction models, including climatic, meteorological and environmental parameters, (iii) implementing follow-up in the form of passive surveillance of animals and humans including recording irregular events (massive abortions in animals, high fever in humans).

To be better prepared for upcoming outbreaks, to be able to predict the introduction of RVF in currently unscathed countries, and to run efficient surveillance programmes, harmonisation and improvements of the diagnostic capacity of countries at risk of RVF are required.

Organising disease-targeted inter-laboratory trials, which should be seen as external quality assessments (EQA) [22–26] is part of the toolbox needed to evaluate the performance of laboratories that routinely diagnose RVF using either serological tests such as ELISAs (enzyme linked immunosorbent assay) or viral genome detection tests.

Here, we report the results of a serological RVF inter-laboratory proficiency test (PT) organised in the framework of the SURE project dealing with surveillance and modelling the risk of transmission of infectious diseases, specifically *peste des petits ruminants* (PPR) and RVF, between the SWIO islands and East and South Africa. A total of 18 laboratories in 13 countries participated in this RVF PT. Among these countries, four are considered to be at risk (Algeria, France (continental and Reunion Island), Morocco, Tunisia), and 10 are considered to be endemic countries for RVF (Madagascar, Mali, Mauritania, France (Mayotte island), Mozambique, Niger, Saudi Arabia, Senegal, South Africa, and Tanzania). This initiative reinforced the interest of laboratories located in at risk or endemic countries which routinely conduct RVF diagnoses to assess and maintain the quality of their RVF diagnostic capacity, which is extremely important when running surveillance programmes at a national or regional scale.

## 2 Materials and methods

### 2.1 Ethics statement

Blood samples were sampled according to national regulations and approved by the regional ethics committee of Languedoc-Roussillon (Comité Régional d'Ethique sur l'Expérimentation Animale- Languedoc-Roussillon), France (approval N˚ APAFIS#14442015081310300000).

### 2.2 Call for participation

In May 2018, an invitation was sent to laboratories in the SWIO region and in countries considered to be at risk of RVFV introduction. A total of 18 laboratories in 13 countries agreed to take part in the PT and were included in the comparative study anonymously (Fig 1).

### 2.3 Preparation of the proficiency test panel

Each participant received a blind coded PT panel of 20 samples following the instructions of the ISO/IEC 17043:2010 as shown in Table 1. The positive samples consisted of bovine sera originating from the RVF infected French overseas department of Mayotte [27]. The negative samples consisted of sera from healthy animals in an RVF free country (mainland France). Each sample was aliquoted and stored at -20 ˚C until shipment.

The panel consisted in 20 samples of non-lyophilised sera including negative (n = 7) and positive sera (n = 11), as well as sera at the limit of detection (n = 2). Each sample was tested with two commercials kits (ID Screen® Rift Valley Fever Competition Multi-species ID.Vet, France and INgezim FVR Compac Ingenasa, Spain). The sera that originally tested positive or doubtful using cELISA (sera numbers 1–10, 16–18 were also tested for the presence of RVF neutralising antibodies using the sero-neutralisation test (SNT), considered as the gold standard by OIE [28] and reference in our PT (Table 1).

All the samples were prepared as 200 µl aliquots for shipment. Each tube in the panel was coded by computer with a random number. Panels were shipped at ambient temperature. No exact recommendation was made regarding the type of test to be used, although the routinely used diagnostic kits were recommended.

### 2.4 Evaluation criteria

The results were interpreted according to the kit manufacturer's instructions.

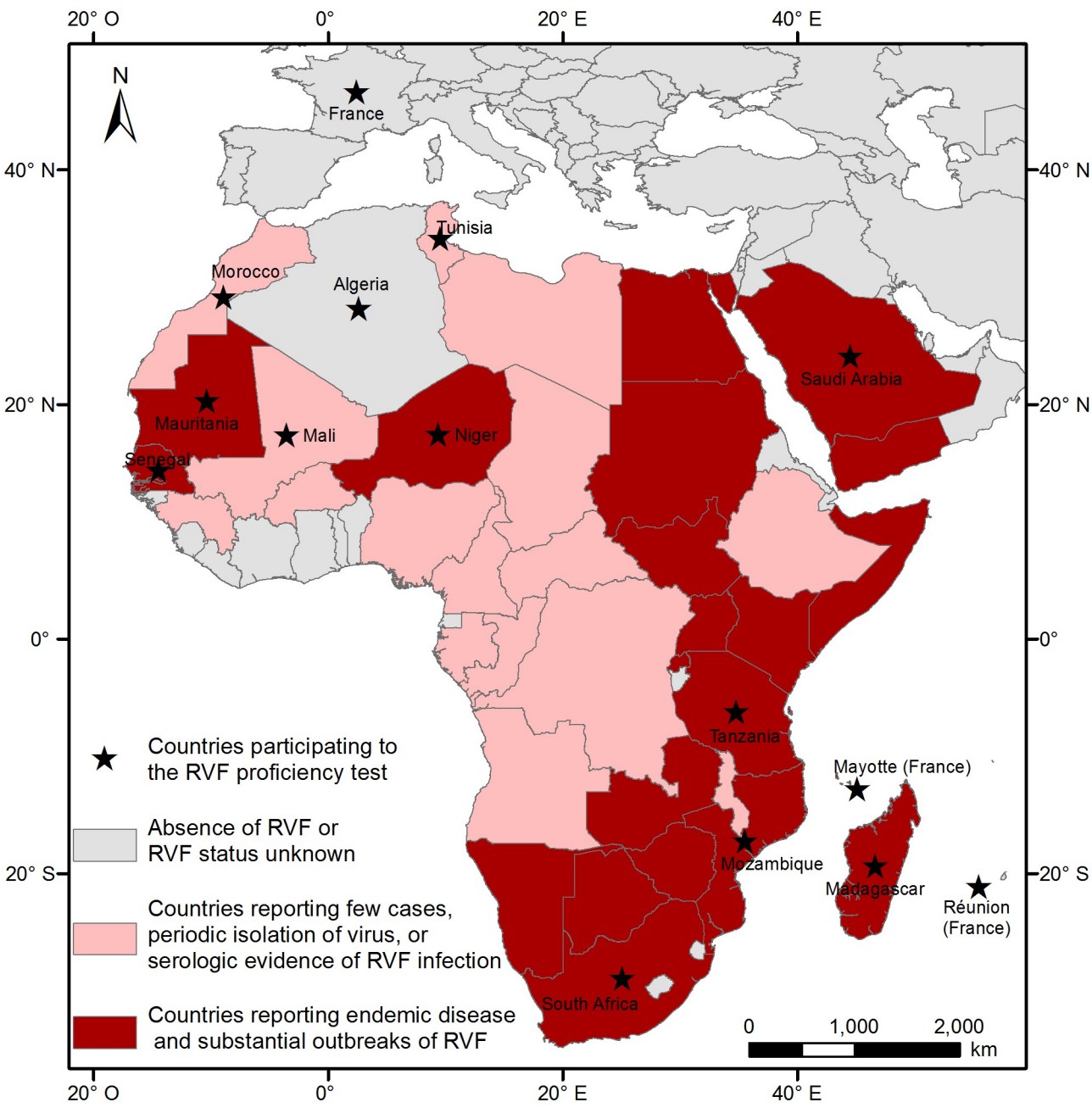

**Fig 1. Countries participating to the RVF proficiency test (PT).**

Results provided by the participants were evaluated based on the following criteria:

- Reporting of results: period not exceeding one month after receipt of the samples.

- Sensitivity: all the expected positive samples are found to be positive.

- Specificity: all the expected negative samples are found to be negative

- Detectability: ID sera samples N˚16 to 20 corresponding to a serial dilution of the ID serum number 8 were used to define the detection threshold for a positive result (the last dilution at

**Table 1. Origin and status of the samples included in the panel used to assess the performance criteria.**

| Serum ID N° | Original animal ID N° | Species | Geographic origin | Date of collection (day/month/year) | Qualitative serological results (gold standard SNT) | Titer (gold standard SNT) | Performance criteria assessed |
|---|---|---|---|---|---|---|---|
| 1 | 1388 | bovine | Mayotte | 15/11/2016 | Positive | 1/30 | Sensitivity |
| 2 | 9568 | bovine | Mayotte | 15/11/2016 | Positive | 1/120 | Sensitivity |
| 3 | 7635 | bovine | Mayotte | 15/11/2016 | Positive | 1/120 | Sensitivity |
| 4 | 7389 | bovine | Mayotte | 15/11/2016 | Positive | 1/100 | Sensitivity |
| 5 | 6228 | bovine | Mayotte | 15/11/2016 | Positive | 1/60 | Sensitivity |
| 6 | 0916 | bovine | Mayotte | 15/11/2016 | Positive | 1/60 | Sensitivity |
| 7 | 4654 | bovine | Mayotte | 15/11/2016 | Positive | 1/100 | Sensitivity |
| 8 | 6449* | bovine | Mayotte | 15/11/2016 | Positive | 1/160 | Repeatability Dose-response relationship |
| 9 | 6449* | bovine | Mayotte | 15/11/2016 | Positive | 1/160 | Repeatability |
| 10 | 6449* | bovine | Mayotte | 15/11/2016 | Positive | 1/160 | Repeatability |
| 11 | 1005 | goat | mainland France | 12/12/2005 | not tested | not tested | Specificity |
| 12 | 1973 | bovine | mainland France | 07/02/2018 | not tested | not tested | Specificity |
| 13 | 9050 | goat | mainland France | 02/12/2005 | not tested | not tested | Specificity |
| 14 | 0444 | bovine | mainland France | 07/02/2018 | not tested | not tested | Specificity |
| 15 | 6025 | goat | mainland France | 25/03/2005 | not tested | not tested | Specificity |
| 16 | Serum ID N°8 in serum ID N°14 (dilution of 1:8) | bovine | Mayotte | 15/11/2016 07/02/2018 | Positive | 1/30 | Dose-response relationship / detectability |
| 17 | Serum ID N°8 diluted in serum ID N°14 (dilution of 1:16) | bovine | Mayotte | 15/11/2016 07/02/2018 | Negative | Negative | Dose-response relationship / detectability |
| 18 | Serum ID N°8 diluted in serum ID N°14 (dilution of 1:32) | bovine | Mayotte | 15/11/2016 07/02/2018 | Negative | Negative | Dose-response relationship Detectability |
| 19 | Serum ID N°8 diluted in serum ID N°14 (dilution of 1:64) | bovine | Mayotte | 15/11/2016 07/02/2018 | Negative | Negative | Dose-response relationship Detectability |
| 20 | Serum ID N°8 diluted in serum ID N° 14 (dilution of 1:128) | bovine | Mayotte | 15/11/2016 07/02/2018 | Negative | Negative | Dose-response relationship Detectability |

ID stands for Identification, N° for Number, SNT stands for Seroneutralisation Test.

* corresponds to a mix of 5 sera numbered 6449, 8932, 128, 8926 and 0082.

which a positive result is accepted), and for a doubtful result (the last dilution at which a doubtful result is accepted).

- Dose-response relationship: a sample run in serial dilutions gives a competition percentage (CP) or a Inhibition percentage (IP) curve depending on the kit used. In the linear portion of the curve, no inverted CP or IP result is accepted. The value of the diluted sample X / 2 must be greater than the CP or IP of the diluted sample X. In addition, the curve should not differ from the expected curve of the candidate laboratory and the set of curves obtained by other laboratories. The competitition percentage (CP) corresponds to the Sample OD value/ Negative control OD value (S/N) x 100) and the inhibition percentage (IP) to 100 –[(Sample OD value/Negative control OD value (S/N) x 100].

- Repeatability: a series of three replicates is included in the panel: ID sera samples N°8, 9 and 10. The qualitative results of these samples should correspond to the expected results

(positive, negative) allowing within-run precision to be evaluated by calculating the mean (m) and standard deviation (SD) of the ELISA test. The deviation (M-m) between the maximum and minimum was calculated for each laboratory. A laboratory with a deviation value of more than the mean m +2 SD value received a notification.

## 3 Results

Eighteen laboratories in 13 countries agreed to take part in the PT and were included in the comparative study anonymously (Fig 1).

### 3.1 Type of tests used

Out of the 18 laboratories which participated in the PT, 14 used the commercially available ID Screen RVF competition multi-species ELISA kit (ID.Vet, France), two laboratories coded N˚ 3, and N˚22 used the INgezim RVF Compac commercially available ELISA kit (Ingenasa, Spain), one laboratory with the coded panels N˚10, and N˚18 used both commercially available kits. The two remaining laboratories coded N˚12 and N˚14 used their own in-house ELISA kits (Table 2).

### 3.2 Global RVF antibody detection

Nineteen data sets were received from the 18 participating laboratories, as a double dataset was received from one laboratory with one dataset for both commercially available ELISA kits, N˚ 10 and N˚18 (ID Screen and INgezim). Table 2 summarises the results, compared to the SNT, considered as the OIE gold standard method and the reference in our PT. The % of correct results is based only on the values obtained for the specificity and sensitivity criteria. The inconclusive status mentioned in the table was interpreted as incorrect when the expected status was positive or negative.

Thirteen out of 14 labs (92.8%) which used the commercially available ID Screen competition RVF multispecies kit reported correct results for all the samples regarding the criteria of sensitivity, repeatability, specificity, and dose-response relationship. The validation criteria of the test conducted by laboratory # 1 were not reached (OD value of the negative control > 0.7) and the results could therefore not be interpreted. The remaining analyses were performed without laboratory # 1.

All three labs (100%) which used the commercially available INgezim RVF compac kit reported correct results for the samples regarding the criteria of repeatability and specificity. Only one of them (1/3, 33%) met the criteria of sensitivity, and none (0/3, 0%) reported expected results for the samples regarding the dose-response relationship criteria.

One out of the two laboratories (lab N˚12) that used its own in-house inhibition ELISA technique reported the expected results for repeatability and specificity but not for sensitivity and the dose-response relationship. The second laboratory (lab #14) that used its own in-house sandwich ELISA technique only reported the expected results for sensitivity, the data reported for the other criteria: repeatability, specificity, and the dose-response relationship, did not correspond to the expected results.

Compared to SNT, the 13 laboratories that used the ID.Vet commercial kit reported 100% correct results (Cohens'Kappa value = 1) whereas a kappa value of 0.88 was reported for the Ingenasa commercial kit used by three laboratories. A kappa value of -0.10 was obtained with the in-house tests and considered inaccepatble.

**Table 2. Results of RVF antibody detection PT.**

| Coded lab | 3 | 18 | 22 | 1 | 2 | 4 | 5 | 6 | 7 | 8 | 9 | 10 | 11 | 13 | 16 | 17 | 19 | 12 | 14 | Expected results | Performance criteria |
|---|---|---|---|---|---|---|---|---|---|---|---|---|---|---|---|---|---|---|---|---|---|
| % of correct results (by lab) | 93 | 93 | 100 | 93 | 100 | 100 | 100 | 100 | 100 | 100 | 100 | 100 | 100 | 100 | 100 | 100 | 100 | 87 | 87 | | |
| D Sens/Spec (%) | 86/100 | 86/100 | 100/100 | 86/100 | 100/100 | 100/100 | 100/100 | 100/100 | 100/100 | 100/100 | 100/100 | 100/100 | 100/100 | 100/100 | 100/100 | 100/100 | 100/100 | 80/100 | 90/80 | | |
| 1 | Pos | Pos | Pos | Pos | Pos | Pos | Pos | Pos | Pos | Pos | Pos | Pos | Pos | Pos | Pos | Pos | Pos | Neg | Pos | Pos | Sensitivity |
| 2 | Pos | Pos | Pos | Pos | Pos | Pos | Pos | Pos | Pos | Pos | Pos | Pos | NT | Pos | Pos | Pos | Pos | Pos | Pos | Pos | Sensitivity |
| 3 | Pos | Pos | Pos | Pos | Pos | Pos | Pos | Pos | Pos | Pos | Pos | Pos | Pos | Pos | Pos | Pos | Pos | Pos | Pos | Pos | Sensitivity |
| 4 | Pos | Pos | Pos | Pos | Pos | Pos | Pos | Pos | Pos | Pos | Pos | Pos | Pos | Pos | Pos | Pos | Pos | Neg | Pos | Pos | Sensitivity |
| 5 | Inc | Inc | Pos | Neg | Pos | Pos | Pos | Pos | Pos | NT | Pos | Pos | Pos | Pos | Pos | Pos | Pos | Pos | Pos | Pos | Sensitivity |
| 6 | Pos | Pos | Pos | Pos | Pos | Pos | Pos | Pos | Pos | Pos | Pos | Pos | Pos | Pos | Pos | Pos | Pos | Pos | Pos | Pos | Sensitivity |
| 7 | Pos | Pos | Pos | Pos | Pos | Pos | Pos | Pos | Pos | Pos | Pos | Pos | Pos | Pos | Pos | Pos | Pos | Pos | Pos | Pos | Sensitivity |
| 8 | Pos | Pos | Pos | Pos | Pos | Pos | Pos | NT | Pos | Pos | Pos | Pos | Pos | Pos | Pos | Pos | Pos | Pos | Pos | Pos | Repeatability + Dose-response relationship |
| 9 | Pos | Pos | Pos | Pos | Pos | Pos | Pos | Pos | Pos | Pos | Pos | Pos | Pos | Pos | Pos | Pos | Pos | Pos | Inc | Pos | Repeatability |
| 10 | Pos | Pos | Pos | Pos | Pos | Pos | Pos | Pos | Pos | Pos | Pos | Pos | Pos | Pos | Pos | Pos | Pos | Pos | Pos | Pos | Repeatability |
| 11 | Neg | Neg | Neg | Neg | Neg | Neg | Neg | Neg | Neg | Neg | Neg | Neg | Neg | Neg | Neg | Neg | Neg | Neg | Neg | Neg | Specificity |
| 12 | Neg | Neg | Neg | Neg | Neg | Neg | Neg | Neg | Neg | Neg | Neg | Neg | Neg | Neg | Neg | Neg | Neg | Neg | Unc | Neg | Specificity |
| 13 | Neg | Neg | Neg | Neg | Neg | Neg | Neg | Neg | Neg | Neg | Neg | Neg | Neg | Neg | Neg | Neg | Neg | Neg | Neg | Neg | Specificity |
| 14 | Neg | Neg | Neg | Neg | Neg | Neg | Neg | Neg | Neg | Neg | Neg | Neg | Neg | Neg | Neg | Neg | Neg | Neg | Neg | Neg | Specificity |
| 15 | NT | Neg | Neg | Neg | Neg | Neg | Neg | Neg | Neg | NT | Neg | Neg | Neg | Neg | Neg | Neg | Neg | Neg | Neg | Neg | Specificity |
| 16 | Neg | Neg | Neg | Neg | Pos | Pos | Pos | Pos | Pos | Pos | Pos | Pos | Pos | Pos | Pos | Pos | Pos | Neg | Pos | Pos | Dose-response relationship |
| 17 | Neg | Neg | Neg | Neg | Inc | Neg | Neg | Pos | Inc | Pos | Neg | Inc | Neg | Inc | Neg | Pos | Neg | Neg | Pos | Neg | Dose-response relationship |
| 18 | Neg | Neg | Neg | Neg | Neg | Neg | Neg | Neg | Neg | Neg | Neg | Neg | Neg | Neg | Neg | Neg | Neg | Neg | Pos | Neg | Dose-response relationship |
| 19 | Neg | Neg | Neg | Neg | Neg | Neg | Neg | Neg | Neg | Neg | Neg | Neg | Neg | Neg | Neg | Neg | Neg | Neg | Neg | Neg | Dose-response relationship |
| 20 | Neg | Neg | Neg | Neg | Neg | Neg | Neg | Neg | Neg | Neg | Neg | Neg | Neg | Neg | Neg | Neg | Neg | Neg | Neg | Neg | Dose-response relationship |
| | INgezim RVF Compac | | | ID Screen RVF competition multi-species kit | | | | | | | | | | | | | | In-house test #1 | In-house test #2 | SNT | |

Neg stands for negative, Pos for positive, Inc for inconclusive, NT for not tested because the tubes were broken or contained no serum upon arrival. SNT stands for sero-neutralisation test, DSens/Spec for diagnostic sensitivity/specificity.

## 3.3 Distribution of RVF PT panel sera depending on the use of the commercially available kits

A total of 17 data sets obtained with commercially available kits were received including 14 datasets for the commercially ID Screen competition RVF multispecies kit, and three datasets for the RVF INgezim Compac kit. Fig 2 shows the distribution of the results depending on the kit used. The 20 sera included in the panel were nicely distinguished by both kits with no overlaps between the negative and the positive status sera.

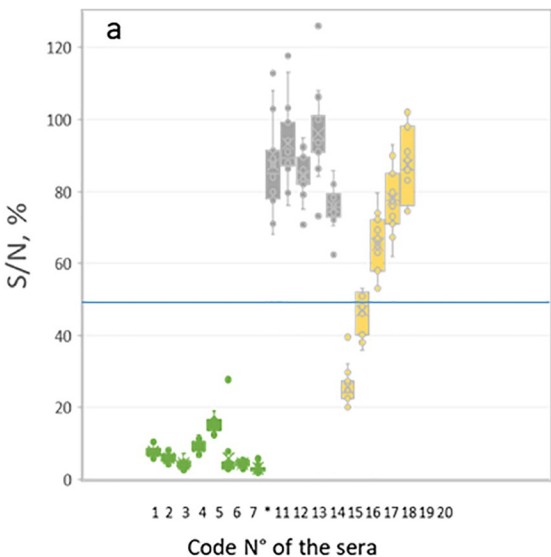
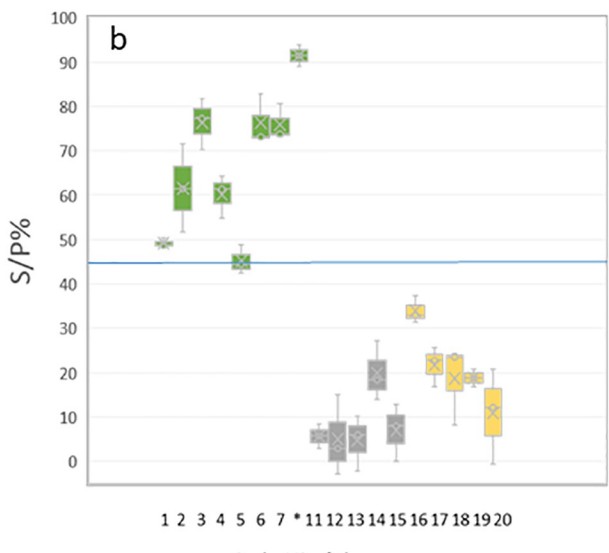

**Fig 2. Serological distribution of the RVF PT panel sera according to the two commercial kits (a) ID Screen® Rift Valley Fever Competition Multi-species and (b) INgezim FVR Compac.** The star * or. indicate the sera N°8 to 10, corresponding to the repeatability criteria.

### 3.4 Dose-response relationship

Six sera samples, numbers 8, 16 to 20, were specifically included in the panel to assess the dose-response relationship criteria. The dose-response relationship was only analysed in the datasets run with the kits that were the most frequently used, the commercially available ID Screen competition RVF multispecies kit and the RVF INgezim Compac kit. Fig 3 shows the dose-response relationship curves for each of the 16 datasets with correct results with no inversion demonstrated by the competition percentage (CP, S/N, %) or inhibition percentage (IP, 100-S/N,%) except for the 2 laboratories N°3 and N°16.

### 3.5 Repeatability

Three sera samples, numbers 8, 9 and 10, were specifically included in the panel to assess the repeatability criteria. Out of the 17 participating laboratories with exploitable results, 13 datasets obtained with the ID Screen® Rift Valley Fever Competition Multi-species kit and 3 with the INgezim FVR Compac kit were analysed and all of them gave satisfactory results except for the lab coded n° 4 (Table 3). The difference between the maximum and the minimum value was considered too high since it was not included between the mean value -2SD (standard deviation) and the mean value +2SD, obtained by all the other participating laboratories. Greater variation was also observed with the ID.Vet kit for repetition number 10 than for the two other sera.

The distribution of the competitition percentage (CP) corresponding to the Sample OD value/Negative control OD value (S/N) x 100) or the inhibition percentage (IP) corresponding to 100 –[(Sample OD value/Negative control OD value (S/N) x 100], using either the ID Screen® Rift Valley fever competition multi-species or the INgezim FVR Compac kits showed tight values, whatever the kit used (Fig 4). Moreover, the median value was identical between the series of values obtained for the three sera in the repetitions, demonstrating good repeatability for both kits used, however the median value obtained using the INgezim compac kit differed more from one series of sera to another than the the ID Screen® Rift Valley Fever Competition Multi-species kit.

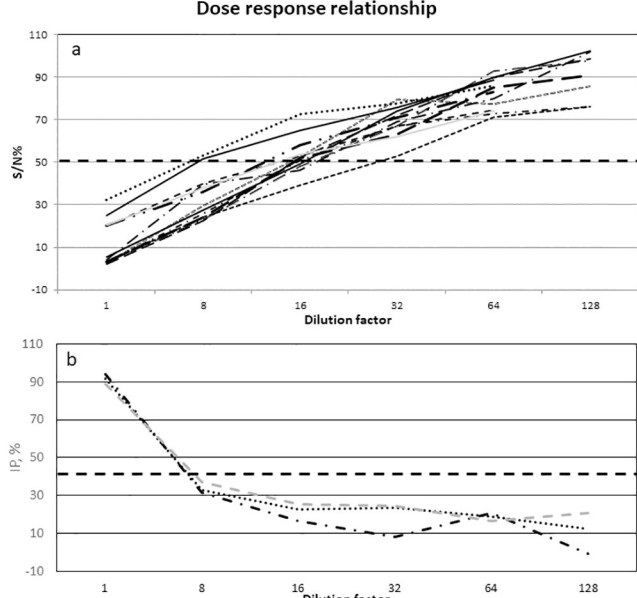

**Fig 3. Dose-response relationship using the commercial ID Screen® Rift Valley Fever Competition multi-species kit (ID.Vet, France).** (a) ID Screen® Rift Valley Fever Competition Multi-species and (b) INgezim FVR Compac. CP stands for competition percentage, IP for Inhibition percentage. Dash lines correspond to the positive threshold.

**Table 3. Repeatability assessed with either the ID Screen® Rift Valley Fever Competition Multi-species or the INgezim FVR Compac kits, based on the 3 sera included in the panel specifically for that purpose.**

| Type of test used | Coded lab N° | Serum N° 8 | Serum N° 9 | Serum N° 10 | Max | Min | M-m | (M-m) mean | SD | Mean +/- 2SD |
|---|---|---|---|---|---|---|---|---|---|---|
| ID Screen® Rift Valley Fever Competition Multi-species (CP, %) | 2 | 4,16 | 4,31 | 4,31 | 4,31 | 4,16 | 0,14 | 0,37 | 0,40 | -0,40/1,17 |
| | 4 | 3,10 | 4,12 | 4,90 | 4,90 | 3,10 | 1,80 | | | |
| | 5 | 2,24 | 2,36 | 2,38 | 2,38 | 2,24 | 0,14 | | | |
| | 6 | NE | 2,00 | 2,00 | 2,00 | 2,00 | 0,00 | | | |
| | 7 | 3,00 | 3,00 | 4,00 | 4,00 | 3,00 | 1,00 | | | |
| | 8 | 3,00 | 3,00 | 3,00 | 3,00 | 3,00 | 0,00 | | | |
| | 9 | 3,00 | 3,00 | 3,00 | 3,00 | 3,00 | 0,00 | | | |
| | 10 | 1,89 | 2,01 | 2,13 | 2,13 | 1,89 | 0,24 | | | |
| | 11 | 3,00 | 3,00 | 3,00 | 3,00 | 3,00 | 0,00 | | | |
| | 13 | 5,28 | 4,95 | 5,71 | 5,71 | 4,95 | 0,75 | | | |
| | 16 | 2,63 | 2,71 | 2,54 | 2,71 | 2,54 | 0,18 | | | |
| | 17 | 2,01 | 2,06 | 1,97 | 2,06 | 1,97 | 0,09 | | | |
| | 19 | 2,83 | 2,31 | 2,60 | 2,83 | 2,31 | 0,52 | | | |
| Ingezim FVR Compac (IP, %) | 3 | 93,85 | 93,63 | 94,07 | 94,07 | 93,63 | 0,44 | 1,19 | 0,50 | 0,19/2,19 |
| | 18 | 91,52 | 90,94 | 89,97 | 91,52 | 89,97 | 1,55 | | | |
| | 22 | 88,92 | 90,50 | 89,85 | 90,50 | 88,92 | 1,58 | | | |

SD stands for standard deviation, N° for number, CP for competition percentage, IP for inhibition percentage.

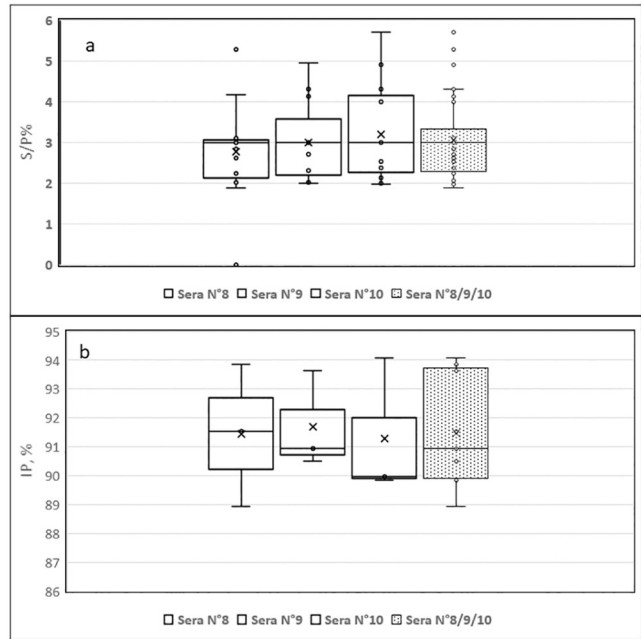

**Fig 4. Assessment of the inter-repeatability based on the 3 sera included in the panel specifically for that purpose.**
Distribution of the Competitition Percentage (CP, %) or Inhibition Percentage (IP, %), depending on the test used, either ID Screen® Rift Valley Fever Competition Multi-species (a) or INgezim FVR Compac (b). White boxes correspond to inter-repeatability for each of the 3 sera whereas the dotted box corresponds to the global inter-repeatability.

As mentioned earlier, of the two laboratories that used in-house techniques, only one (lab #12) met the expected values for the repeatability criteria of whereas the other (lab #14) did not.

## 4 Discussion

New outbreaks of vector-borne and viral diseases in territories or countries where they have never previously been reported, possibly due to climate change, the abundance of newly reported vector species (vector diversity), host or reservoir mobility, reminds us how important it is to maintain technical expertise and to implement routine surveillance programmes of the risk of disease introduction. Among the potentially useful tools for early detection or to predict the introduction of a disease in a disease-free country, is building up laboratory diagnostic capacity able to detect any pathogen that might be introduced into the country as early as possible. External quality assessments are part of the maintenance of the good quality of laboratory diagnosis, specifically for diseases whose differential clinical diagnosis is not easy. The data concerning a serological RVF PT detailed in this paper originated from 13 countries in Africa, the Middle East and the SWIO that are either endemic or at risk of introduction of RVF, reinforcing the ability of these countries to be prepared in the case of RVF emergence.

The use of four different ELISA techniques in this PT underlines the differences in robustness in terms of sensitivity and repeatability between commercially available and in-house kits. In terms of specificity and the dose-response relationship, some of the data obtained were far from what was expected. In this PT, the choice was made to consider the serological status obtained by the SNT method (OIE reference) as the reference status, even though ELISA and SNT methods do not measure the same type of antibodies (anti-N IgG antibodies versus neutralising antibodies). This choice explains the differences in the results obtained for the five

serially diluted panel sera numbered 16 to 20 (performance criteria of dose-response relationship) between the four different techniques. Results may vary from one lab to another, specifically for serum number 17 (serum 8 diluted 1:8) which is just at the limit of detection. The results obtained with the commercial ID Screen® Rift Valley Fever Competition Multi-species detected 100% (13 pos/13 labs) for the panel serum numbered 16 (dilution 1:4), 46% (6 neg/13 labs) for the panel serum numbered 17 (dilution 1:8) which is an acceptable percentage, as the probability to detect the serum as negative or positive is equal and 100% (13 neg/13 labs) for the panel serum numbered 18 (dilution 1:16). This kit has been already evaluated in two other PT with high analytical sensitivity and specificity values [29, 30].

The results obtained by the three labs that use the commercial INgezim FVR Compac kit detected 0% (3 neg/3 labs) for the panel sera numbered 16 (dilution 1:8), 17 (dilution 1:16) and 18 (dilution 1:32) compared to the SNT gold reference status. Even if the findings are only from three laboratories, they demonstrate lower sensitivity than the other commercially available kit and in-house test #2.

Checking additional laboratories which use the two in-house kits is recommended and will surely help to better evaluate their performance criteria, specifically their sensitivity, specificity and dose-relationship response.

The use of a kit that is 100% in accordance with SNT data is recommended. Our RVF serological PT panel only contained sera originating from either bovine or caprine species. Additional sera originating from other species (dromedaries, sheep, and wildlife) needs to be included in future PT panels even though they are not easy to obtain in large quantities. Competitive tests have already been used extensively [31–33], but not sufficiently frequently compared to the SNT gold standard reference.

Finally, the combination of serological (IgM and IgG), molecular methods (PCR) and rapid diagnostic tests (RDT) as point of care (POC) diagnostics in a single external quality assessment (EQA) needs to be strengthened to give policy makers and national or international laboratories a broader view of the sensitivity/specificity of the different techniques, whether commercially available or not. The results of this PT will help choose the appropriate test to be used in each condition (for example, surveillance programmes in endemic countries where viral charges are expected to be low versus active circulation with clinically affected animals mostly harbouring high viral charges).

Among the four types of ELISA tests, only two are commercially available and are currently able to detect specific RVF antibodies. More research is needed to develop alternative tests that could be based on other serological technologies such as multiplex bead based immunoassays (MIA) which have already been developed for RVF and other viral diseases [34–36] and which would help diagnose several diseases at once.

In conclusion, this PT tested the diagnostic capacity of veterinary laboratories in countries neighbouring the EU (North and West Africa) and in trade-related regions in the Indian Ocean (East Africa, Madagascar, SWIO) most of which produced a high percentage of correct results in animal samples taken from naturally infected animals. Taken together, these results show that the countries that took part in this PT are adequately prepared for the early detection of RVF in their respective territories and regions. Building up a regional network will help develop a strategy to better tackle RVF disease in the case of emergence in newly infected territories or huge outbreaks in endemic ones.

## Acknowledgments

We are grateful to Annelise Tran for her help producing the map illustrating the countries that took part in the PT, and to Denise Bastron for registering the participants and sending the

panels to the participating laboratories. The authors would like also to thank the veterinarians (CoopADEM, Food, Agriculture and Forestry Directorate (DAAF, Direction de l'Alimentation, de l'Agriculture et de la Forêt) for their assistance in obtaining the blood samples.

## Author Contributions

**Conceptualization:** Aurélie Pedarrieu, Modou Moustapha Lo.

**Data curation:** Aurélie Pedarrieu, Jean Michel Heraud, Catherine Cetre-Sossah.

**Formal analysis:** Aurélie Pedarrieu, Claudia Filippone, Catherine Cetre-Sossah.

**Investigation:** Aurélie Pedarrieu, Fatiha El Mellouli, Hanane Khallouki, Khalil Zro, Ghizlane Sebbar, Soufien Sghaier, Hafsa Madani, Nadera Bouayed, Mariame Diop, Ahmed Bezeid Ould El Mamy, Yahya Barry, Marthin Dakouo, Abdallah Traore, Haladou Gagara, Maman Moutari Souley, Sara Acha, Laurenco Mapaco, Jelly Chang'a, Denis Nyakilinga, Baratang A. Lubisi, Thabisile Tshabalala, Claudia Filippone, Jean Michel Heraud, Sitty-Bahyat Chamassy, Abdou Achiraffi, Nicolas Keck, Gilda Grard, Kareem Abdelfattah Abdelwahab Mohammed, Abdulwahed Mohammed Alrizqi, Catherine Cetre-Sossah.

**Methodology:** Aurélie Pedarrieu, Jean Michel Heraud, Nicolas Keck, Catherine Cetre-Sossah.

**Project administration:** Catherine Cetre-Sossah.

**Resources:** Aurélie Pedarrieu.

**Supervision:** Catherine Cetre-Sossah.

**Validation:** Aurélie Pedarrieu, Ghizlane Sebbar, Soufien Sghaier, Claudia Filippone, Jean Michel Heraud, Nicolas Keck, Catherine Cetre-Sossah.

**Visualization:** Aurélie Pedarrieu, Fatiha El Mellouli, Hanane Khallouki, Khalil Zro, Ghizlane Sebbar, Soufien Sghaier, Hafsa Madani, Nadera Bouayed, Modou Moustapha Lo, Mariame Diop, Ahmed Bezeid Ould El Mamy, Yahya Barry, Marthin Dakouo, Abdallah Traore, Haladou Gagara, Maman Moutari Souley, Sara Acha, Laurenco Mapaco, Jelly Chang'a, Denis Nyakilinga, Baratang A. Lubisi, Thabisile Tshabalala, Claudia Filippone, Jean Michel Heraud, Sitty-Bahyat Chamassy, Abdou Achiraffi, Gilda Grard, Kareem Abdelfattah Abdelwahab Mohammed, Abdulwahed Mohammed Alrizqi, Catherine Cetre-Sossah.

**Writing – original draft:** Catherine Cetre-Sossah.

**Writing – review & editing:** Aurélie Pedarrieu, Khalil Zro, Soufien Sghaier, Mariame Diop, Marthin Dakouo, Claudia Filippone, Jean Michel Heraud, Nicolas Keck, Catherine Cetre-Sossah.

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
