## [Decision Letter · Decision Letter 0]

11 Mar 2021

PONE-D-21-03249

External quality assessment of Rift Valley fever diagnosis in countries at risk of the disease: African, Indian Ocean and Middle-East regions

PLOS ONE

Dear Dr. Cetre-Sossah,

Thank you for submitting your manuscript to PLOS ONE. After careful consideration, we feel that it has merit but does not fully meet PLOS ONE’s publication criteria as it currently stands. Therefore, we invite you to submit a revised version of the manuscript that addresses the points raised during the review process.

Please explain some confusing information for the counting in the positive or negative sera. As asked include some detailled computation for the sensitivity or  CP/IP. Please also take in account that even if belonging to France, as stated by reviewer 2, the La Réunion Island and Mayotte cannot be classified in the very same geographical context, as there is some differences considering these island either related to Est Africa mainland or to the Indian Ocean island.

We look forward to receiving your revised manuscript.

Kind regards,

Pierre Roques, Ph.D.

Academic Editor

PLOS ONE

Journal Requirements:

'This study was partly funded by the SURE project, Préfecture de la Réunion, INTERREG FEDER TROI 2015-

2017 in the framework of the DP One Health Indian Ocean (www.onehealth-oi.org).'

a. Please provide an amended statement that declares *all* the funding or sources of support (whether external or internal to your organization) received during this study, as detailed online in our guide for authors at http://journals.plos.org/plosone/s/submit-now

Please also include the statement “There was no additional external funding received for this study.” in your updated Funding Statement.

3. We note that Figure 1 in your submission contains map images which may be copyrighted.

We require you to either (a) present written permission from the copyright holder to publish this figure specifically under the CC BY 4.0 license, or (b) remove the figure from your submission:

b. If you are unable to obtain permission from the original copyright holder to publish this figure under the CC BY 4.0 license or if the copyright holder’s requirements are incompatible with the CC BY 4.0 license, please either i) remove the figure or ii) supply a replacement figure that complies with the CC BY 4.0 license. Please check copyright information on all replacement figures and update the figure caption with source information. If applicable, please specify in the figure caption text when a figure is similar but not identical to the original image and is therefore for illustrative purposes only.

Reviewers' comments:

Reviewer's Responses to Questions

**Comments to the Author**

1. Is the manuscript technically sound, and do the data support the conclusions?

Reviewer #1: Partly

Reviewer #2: Yes

2. Has the statistical analysis been performed appropriately and rigorously? 

Reviewer #1: I Don't Know

Reviewer #2: Yes

3. Have the authors made all data underlying the findings in their manuscript fully available?

Reviewer #1: Yes

Reviewer #2: Yes

4. Is the manuscript presented in an intelligible fashion and written in standard English?

Reviewer #1: Yes

Reviewer #2: Yes

5. Review Comments to the Author

Reviewer #1: The aim of this study was to assess the capacity of laboratories to detect antibodies against RVFV in 13 countries, improving the reliability of RVF sero-diagnostics with an inter-laboratory proficiency test. ELISA kit and in house ELISA were used and the specificity and sensitivity of each of them were determined. 20 samples were send for this proficiency test. The status of each sera was determined by the seroneutralisation test considered as the reference. Most of the laboratories find a very good sensitivity and sensitivity. The repeatability of the various ELISA assays was also very high. The dose-response relationship was also followed.

The objectives of this study are clear and de study design is sound. This manuscript needs minor modifications. The most confusing is that authors consider 3 pool of sera, but positive sera are sometime considered in the positive pool, sometime not (ID. 8-10, ID 16-20, specially for the ID 9 -10 - 16)

1 I was confused by the sensitivity calculation. This value must be calculated using independent sera. Here the number of positive independent sera is 8, not 10 (ID 8, 9, 10 are identical and not independent). Please recalculate the sensitivity for each assay.

2 The authors claimed that sera 8 and 16-20 are the same. Why the date of collection vary (15/11/2016 – 07/02/2018). Was the serum collected on two different dates?

3 Table 1

The number of sera and their associated status are highly confusing: 11 are positive but ID8, ID9 and ID10 are the same and must be only considered only in the repeatability test. Then, only 8 sera constitute the positive pool. Seven sera are considered as negative but following the gold standard SNT, I counted 9. Moreover, ID 17 is considered as negative in table2 and did not alter the calculated specificity. Here’s the limit of the SNT can be noticed or the presentation should be changed. Authors cannot change the performance criteria following the sera. Sometime the repeatability assay is included in the results, sometime, dose response result, is not included in the calculation. I recommend that each pool of sera must be considered only for their purpose and not be mixed. The ID of sera at limit of detection must be given for more clarification.

4 table 3

The D sens (%) of the lab1 is false, the Neg (the ID5 being negative) should be inserted for the sensitivity calculation, moreover, the number of positive sera must be 8, not 11 (ELISA pos/ SNT Pos = 7/8). The sensitivity cannot be 100%. Please recalculate sensitivity with 8 positive sera. The authors did not use false positive status (regarding the SNT result) of sera ID 17 in the specificity calculation; I think it is wright if the number of positive sera is 8. I don’t understand the threshold (L206-208) mean + 2SD. (Max-min) is the amplitude of the difference of values, when mean+2SD is a value. The equation could be (max-min)/2 < 2SD ? for both half of the Gaussian. The best criteria must be : Value < Mean +/- 2 SD for the sera.

The ‘SD mean’, ‘SD’ and ‘mean +2 SD’ are calculated for all data. This is not clear as these values are presented in the table. The ‘SD mean’ is the mean? In the legend authors wrote ‘OD values’. Are they really OD value or CP and IP (then %, not OD)

5 L195

The authors must explain briefly in materials and methods how the CP and IP were obtained.

In figure 4 why the authors did not mention the value of ID8 within this graph, as written L196? It should be added if it is the same sera (but because the date of collection is not identical, sample could be different, this must be clarified). The X abscissa must be dilution factor, and log, else, authors must show histogram. The ID of laboratories must be added in the legend, and the positive threshold must be added on the graph.

6 the figure 2 could be removed, because it is non-informative.

7 The figure 3 should be shown in full scale (X abscissa). The positive threshold must be added.

8 Figure 5

Because the sera ID8, ID9 and ID10 are the same, they must be presented in only one box. This figure presented the inter laboratory repeatability, which is different from the intra laboratory repeatability. Authors should divide the two exercises. Then, authors could add and present the qualitative results obtained for all sera. In conclusion, intra laboratory and inter laboratory tests must be clarified.

Reviewer #2: In this study, the authors conducted a serological inter-laboratory proficiency test (PT) to assess the capacity of 18 veterinary laboratories in 13 countries (4 considered to be at risk and 10 to be endemic for RVF) to detect IgG antibodies against RVF. Sixteen laboratories used commercial kits (13 used only the ID screen RVF competition multi-species ELISA kit, 2 used only the INgezim RVF Compac ELISA kit, and 1 used both kits), and 2 used their own in-house ELISA kits. A panel of 20 samples was tested by each laboratory and different criteria were analyzed (sensitivity, specificity, detectability, dose-response relationship, and repeatability). This PT showed that 3 countries need to improve their detection capacities for RVF.

Despite the limitations of the study (described in the discussion), these findings are important from an epidemiologic and public health perspective as efficient diagnostic tools are necessary for the early detection of emerging infectious diseases. Nevertheless, there are minor issues that should be addressed before paper can proceed to publication.

ABSTRACT:

- Lines 54: I suggest adding "the capacity of veterinary laboratories"

INTRODUCTION:

- Line 103: remove parenthesis after “Reunion Island”. Do you consider mainland France and the Reunion Island as one country?

- Lines 102-105: You state that 13 countries participated in the RVF PT. However, 4 countries at risk + 10 countries endemic for RVF = 14 countries. I presume that Mayotte is considered as France but I do not think that it is evident for all readers.

RESULTS:

- Line 189: Are you sure about the kappa value of 0.08 for the in-house tests? Could you provide the detailed calculation?

- Lines 210-212: Details of CP and IP calculation should be previously given at lines 200-201.

DISCUSSION:

- General comment: Since the aim of this kind of study is to improve the diagnostic capacity of laboratories, have the 3 laboratories that provided the worst results planned to use another kit? This should be discussed.

- Line 253: close the parenthesis after "(dilution 1:8"

FIGURES:

Figure 1: I suggest writing on the map the name of the countries and the code of the labs participating in the RVF PT.

In the text at line 113, it is indicated that 13 countries participated in the RVF PT whereas on the map 15 stars are represented. I suppose that Mayotte and Reunion islands are considered as belonging to France. Consequently I suggest indicating on the map "Mayotte (France)" and "Reunion (France)".

Figure 2: Information given in the Figure 2 are already given in the Table 2. I suggest removing this figure and replacing in the text "(Fig 2)" by "(Table 2)".

6. PLOS authors have the option to publish the peer review history of their article (what does this mean?). If published, this will include your full peer review and any attached files.

Reviewer #1: No

Reviewer #2: No

---

## [Author Response · Author response to Decision Letter 0]

16 Apr 2021

PONE-D-21-03249

Dear Editor and reviewers,

We thank you for your constructive and encouraging comments about our manuscript entitled « External quality assessment of Rift Valley fever diagnosis in countries at risk of the disease: African, Indian Ocean and Middle-East regions », as well as for giving us an opportunity to improve our manuscript further by addressing them. 

On behalf of all co-authors, please find below our answers to each comment, and details of the amendments. The changes are shown in the revised manuscript using “Track changes” using line numbers with track changes on. 

The Funding Statement is now as follows : 'This study was partly funded by the SURE project, Préfecture de la Réunion, INTERREG FEDER TROI 2015-

2017 in the framework of the DP One Health Indian Ocean (www.onehealth-oi.org). There was no additional external funding received for this study.” 

Mayotte and Reunion islands are now mentioned as belonging to France in the Figure 1 and in the Material and Methods section. Sensitivity and specificity performance criteria have been recalculated as asked by the reviewers by considering only individual serum ant not pooled sera.

This material has not and will not be offered elsewhere for possible publication, as long as it is under consideration by PLOS One.

Sincerely yours,

Catherine CETRE-SOSSAH

Responses to the reviewers’ comments

Comment 1 : Thank you for stating in your Funding Statement: 'This study was partly funded by the SURE project, Préfecture de la Réunion, INTERREG FEDER TROI 2015-

2017 in the framework of the DP One Health Indian Ocean (www.onehealth-oi.org).'

Response 1 : The funding statement is modified accordingly and is now as follows : 'This study was partly funded by the SURE project, Préfecture de la Réunion, INTERREG FEDER TROI 2015-2017 in the framework of the DP One Health Indian Ocean (www.onehealth-oi.org). There was no additional external funding received for this study. (line 298-301)” 

Comment 2 : We note that Figure 1 in your submission contains map images which may be copyrighted. All PLOS content is published under the Creative Commons Attribution License (CC BY 4.0), which means that the manuscript, images, and Supporting Information files will be freely available online, and any third party is permitted to access, download, copy, distribute, and use these materials in any way, even commercially, with proper attribution. For these reasons, we cannot publish previously copyrighted maps or satellite images created using proprietary data, such as Google software (Google Maps, Street View, and Earth). For more information, see our copyright guidelines: http://journals.plos.org/plosone/s/licenses-and-copyright.

We require you to either (a) present written permission from the copyright holder to publish this figure specifically under the CC BY 4.0 license, or (b) remove the figure from your submission:

“I request permission for the open-access journal PLOS ONE to publish XXX under the Creative Commons Attribution License (CCAL) CC BY 4.0 (http://creativecommons.org/licenses/by/4.0/). Please be aware that this license allows unrestricted use and distribution, even commercially, by third parties. Please reply and provide explicit written permission to publish XXX under a CC BY license and complete the attached form.”Please upload the completed Content Permission Form or other proof of granted permissions as an "Other" file with your submission. In the figure caption of the copyrighted figure, please include the following text: “Reprinted from [ref] under a CC BY license, with permission from [name of publisher], original copyright [original copyright year].”

 b. If you are unable to obtain permission from the original copyright holder to publish this figure under the CC BY 4.0 license or if the copyright holder’s requirements are incompatible with the CC BY 4.0 license, please either i) remove the figure or ii) supply a replacement figure that complies with the CC BY 4.0 license. Please check copyright information on all replacement figures and update the figure caption with source information. If applicable, please specify in the figure caption text when a figure is similar but not identical to the original image and is therefore for illustrative purposes only.

Response 2 : As authors of Figure 1, we agree to publish the Figure under the CC BY 4.0 license.

Comment 3 : Your ethics statement should only appear in the Methods section of your manuscript. If your ethics statement is written in any section besides the Methods, please move it to the Methods section and delete it from any other section. Please ensure that your ethics statement is included in your manuscript, as the ethics statement entered into the online submission form will not be published alongside your manuscript.

Response 3 : The manuscript is modified accordingly (Line 112-115)

Reviewer #1: The aim of this study was to assess the capacity of laboratories to detect antibodies against RVFV in 13 countries, improving the reliability of RVF sero-diagnostics with an inter-laboratory proficiency test. ELISA kit and in house ELISA were used and the specificity and sensitivity of each of them were determined. 20 samples were send for this proficiency test. The status of each sera was determined by the seroneutralisation test considered as the reference. Most of the laboratories find a very good sensitivity and sensitivity. The repeatability of the various ELISA assays was also very high. The dose-response relationship was also followed.

Comment 4 :The objectives of this study are clear and de study design is sound. This manuscript needs minor modifications. The most confusing is that authors consider 3 pool of sera, but positive sera are sometime considered in the positive pool, sometime not (ID. 8-10, ID 16-20, specially for the ID 9 -10 - 16).

Response 4 : The authors included in the PT panel a total of 20 sera of different origins with 2 pools of mixed sera, a first pool ID8-ID9-ID10 originating from a mix of 5 sera, and a second pool ID16-ID17-ID18-ID19-ID20 originating from ID8 (a mix of 5 sera) diluted in ID14. The second pool ID16-ID17-ID18-ID19-ID20, as mentioned in Table 1, corresponds to a serial dilution of positive mixed sera (ID8) in a RVF negative serum (ID14) starting from the dilution 1:8 (ID16) detected positive to a dilution 1:128 (ID20) detected negative including intermediate dilutions detected negative as well (dilution 1:16 (ID17), dilution 1:32 (ID18), dilution 1:64 (ID19). 

Comment 5 : I was confused by the sensitivity calculation. This value must be calculated using independent sera. Here the number of positive independent sera is 8, not 10 (ID 8, 9, 10 are identical and not independent). Please recalculate the sensitivity for each assay.

Response 5 : The authors agree with the reviewer. Only independent sera are now taken into account for the sensitivity calculation, with 7 sera (ID 1 to ID7) instead of 10 sera (ID1 to ID10) in the initial version of the manuscript. Sensitivity has been recalculated. The ciriteria of sensitivity is deleted for ID8-ID9-ID10. Table 1 and Table 2 are modified accordingly.

Comment 6 : The authors claimed that sera 8 and 16-20 are the same. Why the date of collection vary (15/11/2016 – 07/02/2018). Was the serum collected on two different dates?

Response 6 :The authors agree with the reviewer. The 5 sera included in the ID8 were collected 15/11/2016, and the serum ID14 was collected the 07/02/2018. The authors corrected the 5 lines of Table 1 corresponding to ID16 to ID20 by adding the additional date of collection of ID8, which is 15/11/2016.

Comment 7 : Table 1. The number of sera and their associated status are highly confusing: 11 are positive but ID8, ID9 and ID10 are the same and must be only considered only in the repeatability test. Then, only 8 sera constitute the positive pool. Seven sera are considered as negative but following the gold standard SNT, I counted 9. Moreover, ID 17 is considered as negative in table2 and did not alter the calculated specificity. Here’s the limit of the SNT can be noticed or the presentation should be changed. Authors cannot change the performance criteria following the sera. Sometime the repeatability assay is included in the results, sometime, dose response result, is not included in the calculation. I recommend that each pool of sera must be considered only for their purpose and not be mixed. The ID of sera at limit of detection must be given for more clarification.

Response 7 : The authors agree with the reviewer. Only independent sera were taken into account 

- The 3 sera ID8-ID9-ID10 were only used for the repeatability criteria. 

- Sensitivity calculation : 7 sera (ID 1 to ID7) were used for the sensitivity calculation instead of 10 sera (ID1 to ID10) as mentioned in the initial version of the manuscript. The authors did not want to include ID16 as it is a pool of sera. The values of Sens are modified in Table 2 accordingly

- Specificity calculation : 5 sera (ID11 to ID15) were used for the specificity calculation. It is not possible to include the 4 extra sera proposed by the reviewer (ID17 to ID20) since they are a pool of sera, and we only consider individual serum for criteria performance calculations.

Comment 8 : table 3. The D sens (%) of the lab1 is false, the Neg (the ID5 being negative) should be inserted for the sensitivity calculation, moreover, the number of positive sera must be 8, not 11 (ELISA pos/ SNT Pos = 7/8). The sensitivity cannot be 100%. Please recalculate sensitivity with 8 positive sera. The authors did not use false positive status (regarding the SNT result) of sera ID 17 in the specificity calculation; I think it is right if the number of positive sera is 8.

Response 8 : The authors agree with the reviewer. The Sens of Lab N°1 was false, it is now changed in Table 2, the value of 100 is now 86. Table 1 is modified accordingly. Sensitivity calculation : 7 sera (ID 1 to ID7) were used for the sensitivity calculation instead of 10 sera (ID1 to ID10) as mentioned in the initial version of the manuscript. The authors did not want to include ID16 as it is a pool of sera. The values of Sens are modified in Table 2 accordingly

Comment 8bis : Table 3. I don’t understand the threshold (L206-208) mean + 2SD. (Max-min) is the amplitude of the difference of values, when mean+2SD is a value. The equation could be (max-min)/2 < 2SD ? for both half of the Gaussian. The best criteria must be : Value < Mean +/- 2 SD for the sera.

The ‘SD mean’, ‘SD’ and ‘mean +2 SD’ are calculated for all data. This is not clear as these values are presented in the table. The ‘SD mean’ is the mean? In the legend authors wrote ‘OD values’. Are they really OD value or CP and IP (then %, not OD).

Response 8bis: the authors took into consideration the criteria : Value < Mean +/- 2 SD for the sera. The SD mean is replaced by (M-m) mean. Table 3 is modified accordingly. « OD values » in the legend is deleted. CP and IP, % are added in the legend of table 3.

Comment 9. L195.The authors must explain briefly in materials and methods how the CP and IP were obtained. In figure 4 why the authors did not mention the value of ID8 within this graph, as written L196? It should be added if it is the same sera (but because the date of collection is not identical, sample could be different, this must be clarified). The X abscissa must be dilution factor, and log, else, authors must show histogram. The ID of laboratories must be added in the legend, and the positive threshold must be added on the graph.

Response 9 : The following sentences have been added in the paragraph dealing with the dose response relationship in the Material and Methods section Lines 149-155. It is now : “Dose-response relationship : a sample run in serial dilutions gives a competition percentage (CP) or a Inhibition percentage (IP) curve depending on the kit used. In the linear portion of the curve, no inverted CP or IP result is accepted. The value of the diluted sample X / 2 must be greater than the CP or IP of the diluted sample X. In addition, the curve should not differ from the expected curve of the candidate laboratory and the set of curves obtained by other laboratories. The competitition percentage (CP) corresponds to the Sample OD value/Negative control OD value (S/N) x 100) and the inhibition percentage (IP) to 100 – [(Sample OD value/Negative control OD value (S/N) x 100]”. By mistake, ID8 was not present in the figure 4 even though it the stock serum that was serially diluted from 1:8 to 1:128 (ID16 to ID20). The X axis is changed with dilution factor and in log function (1 to 128) instead of sera ID numbers. 

A positive threshold is now added on the graph Initial Figure 4 is modified accordingly and is now Figure 3 (Figure 2 of the manuscript being removed). The ID of laboratories can not be added in the legend as this comparative study is the result of participating laboratories willing to participate anonymously to this study. 

Comment 10 : the figure 2 could be removed, because it is non-informative.

Response 10 : The figure 2 has been removed.

Comment 11 : The figure 3 should be shown in full scale (X abscissa). The positive threshold must be added.

Response 11 : Figure 3 is Figure 2 in the revised version with the addition of the positive threshold and the full scale of the X axis.

Comment 12 : Figure 5. Because the sera ID8, ID9 and ID10 are the same, they must be presented in only one box. This figure presented the inter laboratory repeatability, which is different from the intra laboratory repeatability. Authors should divide the two exercises. Then, authors could add and present the qualitative results obtained for all sera. In conclusion, intra laboratory and inter laboratory tests must be clarified.

Response 12 :The authors agree that the figure 5 illustrates the inter laboratory repeatability, the title of the figure is changed accordingly. This figure shows the behaviour of each of the 3 sera independantly. An additionnal grey box is added including the values of all the laboratories for the 3 sera all mixed together representing the global inter-repeatability of the PT for each kit. The intra-repeatability is also illustrated in an horizontal reading in Table 3, with the min and Max values obtained for each of the 3 sera used for repeatability assessment (ID8-ID9-ID10). Figure 5 (figure 4 in the revised manuscript) is modified accordingly.

Reviewer #2: 

 Comment 13 : ABSTRACT:- Lines 54: I suggest adding "the capacity of veterinary laboratories"

Response 13 : The manuscript is modified accordingly line 54.

Comment 14 : INTRODUCTION: - Line 103: remove parenthesis after “Reunion Island”. Do you consider mainland France and the Reunion Island as one country?

Response 14 : Reunion and Mayotte islands even if located in the Indian Ocean are two French overseas departments. For a better undertsanding, the sentences are modified as follows Lines 102-105: A total of 18 laboratories in 13 countries participated in this RVF PT. Among these countries, four are considered to be at risk (Algeria, France (continental and Reunion Island), Morocco, Tunisia), and 10 are considered to be endemic countries for RVF (Madagascar, Mali, Mauritania, France (Mayotte island), Mozambique, Niger, Saudi Arabia, Senegal, South Africa, and Tanzania).

Comment 15 : Lines 102-105: You state that 13 countries participated in the RVF PT. However, 4 countries at risk + 10 countries endemic for RVF = 14 countries. I presume that Mayotte is considered as France but I do not think that it is evident for all readers.

Response 15 : The sentence is modified as follows Line 102-105: A total of 18 laboratories in 13 countries participated in this RVF PT. Among these countries, four are considered to be at risk (Algeria, France (continental and Reunion Island), Morocco, Tunisia), and 10 are considered to be endemic countries for RVF (Madagascar, Mali, Mauritania, France (Mayotte island), Mozambique, Niger, Saudi Arabia, Senegal, South Africa, and Tanzania).

Comment 16 : RESULTS:- Line 189: Are you sure about the kappa value of 0.08 for the in-house tests? Could you provide the detailed calculation?

Response 16: The calculation of the kappa value was done in comparison to SNT results considered as gold standard. The results obtained for each serum by each test were converted in proportions, which allowed their comparison between tests. As recommended by reviewer 1, the 3 sera used for repeatability (ID8-ID9-ID10) were not used for the assessment for sensitivity. The kappa values changed. Text is changed accordingly as follows : « Compared to SNT, the 13 laboratories that used the ID.Vet commercial kit reported 100% correct results (Cohens’Kappa value =1) whereas a kappa value of 0.88 was reported for the Ingenasa commercial kit used by three laboratories. A kappa value of -0.10 was obtained with the in-house tests and considered inacceptable ».

Comment 17 : Lines 210-212: Details of CP and IP calculation should be previously given at lines 200-201.

Response 17: The following sentences have been added Lines 153-155. The competitition percentage (CP) corresponds to the Sample OD value/Negative control OD value (S/N) x 100) and the inhibition percentage (IP) to 100 – [(Sample OD value/Negative control OD value (S/N) x 100.

Comment 18 : DISCUSSION:- General comment: Since the aim of this kind of study is to improve the diagnostic capacity of laboratories, have the 3 laboratories that provided the worst results planned to use another kit? This should be discussed.

Response 18: A new sentence is added line 268 « The use of a kit that is 100% in accordance with SNT data is recommended ».

Comment 19 : Line 253: close the parenthesis after "(dilution 1:8"

Response 19: the manuscript is modified accordingly

Comment 20 : Figure 1: I suggest writing on the map the name of the countries and the code of the labs participating in the RVF PT. In the text at line 113, it is indicated that 13 countries participated in the RVF PT whereas on the map 15 stars are represented. I suppose that Mayotte and Reunion islands are considered as belonging to France. Consequently I suggest indicating on the map "

Response 20: the authors took into consideration the comment regarding the addition of the names of the country on the map with Mayotte (France) and Reunion (France) as both islands are considered as part of France. The authors did not write the code of the laboratories on the map as the 18 laboratories that agreed to take part in the PT were included in the comparative study anonymously.

Comment 21 : Figure 2: Information given in the Figure 2 are already given in the Table 2. I suggest removing this figure and replacing in the text "(Fig 2)" by "(Table 2)".

Response 21: the manuscript is modified accordingly (line 170)________________________________________

---

## [Decision Letter · Decision Letter 1]

23 Apr 2021

External quality assessment of Rift Valley fever diagnosis in countries at risk of the disease: African, Indian Ocean and Middle-East regions

PONE-D-21-03249R1

Dear Dr. Cetre-Sossah,

We’re pleased to inform you that your manuscript has been judged scientifically suitable for publication and will be formally accepted for publication once it meets all outstanding technical requirements.

Kind regards,

Pierre Roques, Ph.D.

Academic Editor

PLOS ONE

Additional Editor Comments (optional):

Reviewers' comments:

Reviewer's Responses to Questions

**Comments to the Author**

1. If the authors have adequately addressed your comments raised in a previous round of review and you feel that this manuscript is now acceptable for publication, you may indicate that here to bypass the “Comments to the Author” section, enter your conflict of interest statement in the “Confidential to Editor” section, and submit your "Accept" recommendation.

Reviewer #1: All comments have been addressed

Reviewer #2: All comments have been addressed

2. Is the manuscript technically sound, and do the data support the conclusions?

Reviewer #1: Yes

Reviewer #2: Yes

3. Has the statistical analysis been performed appropriately and rigorously? 

Reviewer #1: Yes

Reviewer #2: N/A

4. Have the authors made all data underlying the findings in their manuscript fully available?

Reviewer #1: Yes

Reviewer #2: Yes

5. Is the manuscript presented in an intelligible fashion and written in standard English?

Reviewer #1: Yes

Reviewer #2: Yes

6. Review Comments to the Author

Reviewer #1: (No Response)

Reviewer #2: (No Response)

7. PLOS authors have the option to publish the peer review history of their article (what does this mean?). If published, this will include your full peer review and any attached files.

Reviewer #1: No

Reviewer #2: No

---

## [Editor Report · Acceptance letter]

27 Apr 2021

PONE-D-21-03249R1 

External quality assessment of Rift Valley fever diagnosis in countries at risk of the disease: African, Indian Ocean and Middle-East regions 

Dear Dr. Cetre-Sossah:

I'm pleased to inform you that your manuscript has been deemed suitable for publication in PLOS ONE. Congratulations! Your manuscript is now with our production department. 

Kind regards, 

on behalf of

Dr. Pierre Roques 

Academic Editor

PLOS ONE